# Valorization of Bayer Red Mud in a Circular Economy Process: Valuable Metals Recovery and Further Brick Manufacture

Carlos Leiva ⬡, Fátima Arroyo-Torralvo *⬡, Yolanda Luna-Galiano, Rosario Villegas, Luis Francisco Vilches and Constantino Fernández Pereira

Departamento de Ingeniería Química y Ambiental, Escuela Técnica Superior de Ingeniería, Universidad de Sevilla, Camino de los Descubrimientos s/n, 41092 Sevilla, Spain
* Correspondence: fatimarroyo@us.es; Tel.: +34-9544-82274

**Abstract:** In this work, the recovery of valuable metals from Bayer red mud using hydrometallurgical techniques and the subsequent use of the solid remaining after leaching as the principal component of the fired bricks were analyzed. Water, sulfuric acid, and sodium hydroxide were used as leaching agents. Different L/S ratios and contact times were also tested. According to technical, economic, and environmental considerations, the optimal conditions to recover valuable elements from red mud were 2 M $H_2SO_4$, in contact for 24 h, with an L/S ratio = 5. Under these conditions, high leaching yields of valuable elements such as La (47.6%) or V (11%) were achieved. After the leaching process, the remaining solid was mixed with clay and water to produce bricks. Two doses of red mud (50 and 80% w) and two different sintering temperatures (900 and 1100 °C) were tested. When the proportion of treated RM in the mix was increased, the compressive strength of the bricks was reduced, but it was increased as the sintering temperature was increased. The environmental safety of the bricks manufactured (leaching of heavy metals and radionuclides) was also studied, and it was found that it was more favorable when red mud was treated instead of fresh red mud being used.

**Keywords:** red mud; circular economy; hydrometallurgy; fired brick; natural radionuclides; heavy metals

## 1. Introduction

Red mud (RM) is a hazardous waste left over from the Bayer process, in which alumina is extracted from bauxite by caustic digestion. Red mud is a mixture of the original bauxite components and other components formed or introduced by means of the Bayer process. In the process, it is necessary to use 1.9–3.6 tons of bauxite to produce 1 ton of alumina, depending on the mineral quality. The annual production of alumina in 2020 was 133 MM tons, which represents the generation of 175 MM tons of red mud [1]. As RM is considered a hazardous residue [2] due to its toxic/heavy and radioactive metal content and its high alkalinity, it is necessary to develop strategies to manage this product in a way that respects the environment.

Recycling of RM has been studied previously in an attempt to find new applications for it in different sectors [3]. Red mud is also rich in different valuable metals, such as Fe, Ti, Al, Si, and Na as major elements; Mg, Ca, Mn, and V as minor elements [3]; and rare-earth elements [4].

Red mud has been widely studied for its application in building materials [4] for/in the preparation of cement and concrete [5], since red mud contains $SiO_2$, $Al_2O_3$, and CaO, which could replace a portion of clay in cement and concrete. In addition, red mud has also been studied as raw material for glass in ceramic materials [6] and in the preparation of geopolymers [4]. In this last application, geopolymers could immobilize radioactive elements of red mud [7].

Metal oxides such as $Al_2O_3$, $Fe_2O_3$, and $TiO_2$ from RM are also used for the preparation of catalysts; many research works have studied the utilization of RM in different catalytic

processes such as hydrogenation, transesterification, pyrolysis, and oxidation [4]. Red mud could also be used in adsorption processes due to its interesting specific surface area and fine particle size [8]. Thus, some works have been carried out to investigate the potential of RM in water treatment to remove heavy metals and other inorganic, organic, and biological contaminants [9,10]. Other studies have described its potential in gas treatment to fix carbon dioxide, nitride, sulfur, and fluoride compounds [4,11].

Fe, Al, Si, and Ti can be recovered by pyrometallurgical methods [3]. Hydrometallurgical methods can also be used to recover valuable components [4]; acid solutions (sulfuric, hydrochloric, and nitric), organic solutions (citric, acetic), ionic liquids, carbonate reagents, and bioleaching and alkaline reagents can be used to recover Fe, Al, Ti, Zr, Si, Ga, Sc, and rare-earth elements, etc. Magnetic separation is another technology studied to separate iron from RM [4]. Combined technologies (pyrometallurgical, hydrometallurgical, and magnetic) have also been used to recover Fe, Al, Ti, and Si [12]. Biological processes using fungi and bacteria have also been evaluated [13].

The aim of this work is to turn waste management into resource management. The idea that waste can be a source of raw materials or energy is not new. In all previously published valorization routes, RM has been proposed as a construction material or as a source of valuable metals; however, both routes seem mutually exclusive. In the present work, both routes are joined and studied together: first, the potential use of RM as a source of different elements released after acid attack and leaching, and subsequently the potential valorization of the leached RM as a construction material used for brick manufacturing, analyzing the implications of the extraction process on the properties of the red mud as components of bricks. This valorization strategy would have a double profit; on the one hand, the RM disposal problem could be reduced, and on the other hand, the recovery of valuable metals and the use of leached RM in brick formulations could produce a saving of raw materials, which could involve economic and environmental benefits.

## 2. Materials and Methods

### 2.1. Characterization of Red Mud and Clay

RM from a Bayer process and clay (CL) were used. The major components of RM and CL have been published elsewhere [14]. Red mud contains a lower value of $SiO_2$ than CL (4.87% vs. 75.66 wt%). The contents of $Al_2O_3$ in RM are higher than CL (18.08 vs. 11.25 wt%). The $TiO_2$ content of red mud is also high (9.33 wt%), while CL has a negligible titanium content.

To analyze the possible recovery of the valuable elements of RM, a complete chemical characterization was performed that included minor components (Table 1). The major and trace components were performed by X-ray fluorescence spectroscopy analysis (Bruker AXS GmbH, Karlsruhe, Germany) at the Research, Technology, and Innovation Center at the University of Seville (CITIUS).

**Table 1.** Minor components of red mud.

| Components | mg/kg | Components | mg/kg | Components | mg/kg |
|---|---|---|---|---|---|
| As | 63.9 | Mo | 12.3 | **Ta** | **7.0** |
| Ba | 193.7 | **Nb** | **161.2** | Th | 121.8 |
| Br | 4.0 | Nd | 84.1 | U | 13.0 |
| Ce | 259.0 | Ni | 17.3 | **V** | **1169** |
| Cr | 1818.0 | P | 1210 | W | 48.2 |
| Cu | 74.9 | Pb | 62.0 | Y | 115.8 |
| **Ga** | **73.4** | Sb | 5.6 | Zn | 46.0 |
| **Hf** | **20.2** | **Sc** | **57.9** | Zr | 2117 |
| **In** | **14.7** | Se | 2.0 | F | 2118 |
| **La** | **112.6** | Sn | 9.3 | S | 514 |
| Mn | 276.4 | Sr | 93.9 | | |
| Ag, Bi, Cd, Cl, Co, Cs, Ge, Hg, Rb, Sm, and Te | | | | non-detected | |

On the one hand, RM contained some heavy metals, such as chromium, barium, and vanadium, which could cause some leaching problems, and thorium which could cause some radiological problems. On the other hand, RM contained many elements included in the fourth (last) EU list of Critical Raw Materials [15]. In some cases, the contents of Ti, Ta, Ga, V, or La were considerably high and were like those reported by other authors [4]. In Table 1 the Critical Raw Materials are written in bold.

### 2.2. Red Mud Valorization by Leaching

Leaching experiments using aqueous solutions of $H_2SO_4$ and NaOH were tested. The leaching tests were performed in two stages: first, a screening step was carried out to determine the extraction capacity of each leaching agent, and in a second step the leaching conditions were optimized. Table 2 summarizes the test conditions carried out in the first and second stages. The influence of leaching time and the L/S ratio on the metal recovery from RM was evaluated using an experimental design. For this purpose, a two-factor factorial design was used that included $4 \times 3$ experiments.

**Table 2.** Leaching tests carried out in stages 1 and 2.

| Screening Tests (Contact Time = 24 h and L/S = 10) | |
| --- | --- |
| Leaching agents | Concentration |
| Water | - |
| Sulfuric Acid | 0.5 M, 1 M, 2 M |
| Sodium hydroxide | 0.5 M, 1 M, 2 M |
| Optimization of leaching operating conditions (2 M $H_2SO_4$) | |
| L/S ratio | 2, 5, 10 L/kg |
| Contact time | 1, 2, 6, 24 h |

The use of $H_2SO_4$ as a leaching agent to recover different elements of RM has been previously described [16,17]. Sulfuric acid concentrations were selected based on the results of Cui et al. [18], which showed a high extraction yield of REE at room temperature when the $H_2SO_4$ concentration was greater than 1 N (0.5 M). The use of NaOH as a leaching solution to valorize RM has also been previously described [18,19]. NaOH concentrations were based on a study performed by Borra et al. [16], which showed that at alkaline pH, the minimum leaching yield of some toxic elements was achieved at a pH around 13.

The experimental extraction yields were fitted by a mathematical model using Stat-Ease Design-Expert software (Version 10, Minneapolis, MN, USA), and the analysis of variance was used for statistical analysis. The quality of fit of the polynomial model was expressed by $R^2$, and its statistical significance was examined using the F test. The experiments were carried out in duplicate employing 100 g of RM. After contact, the liquid and solid phases were separated for chemical analyses.

### 2.3. Brick Manufacture

The brick raw materials (red mud-treated (RM-T), clay (CL), and water) were first mixed for 4 min. Subsequently, cylindrical molds were filled with wet pastes and compressed at 5 MPa for five minutes. The characteristics of molding spectrum pressures were between 4 and 50 MPa [20]. Finally, the specimens were cured at 25 °C for 48 h, followed by drying at 100 °C for two days. Table 3 shows the proportions of RM-T and CL, and the water/solid ratio. Two firing temperatures (900 and 1100 °C) were tested. As can be seen in Table 3, the water/solid ratio was higher as the RM-T content increased to obtain a fluid mixture.

**Table 3.** Brick compositions (wt%).

|             | CL  | RM-T | Water/Solid Ratio |
|-------------|-----|------|-------------------|
| CL-900      | 100 | 0    | 0.12              |
| RM-T-50-900 | 50  | 50   | 0.30              |
| RM-T-80-900 | 20  | 80   | 0.35              |
| CL-1100     | 100 | 0    | 0.12              |
| RM-T-50-1100| 50  | 50   | 0.30              |
| RM-T-80-1100| 20  | 80   | 0.35              |

### 2.4. Heating Method for Firing Bricks

Different heating phases (Figure 1) were used: (A) from 25 °C to 500 °C at 100 °C/h, (B) from 500 °C to the final temperature (900 °C or 1100 °C) at 50 °C/h, and (C) keeping the final temperature constant for 8 h and then cooling to room temperature. The two heating methods were selected because they were used in a previous work, with red mud without a previous leaching process, in order to analyze the difference between the recycling of both red muds [14].

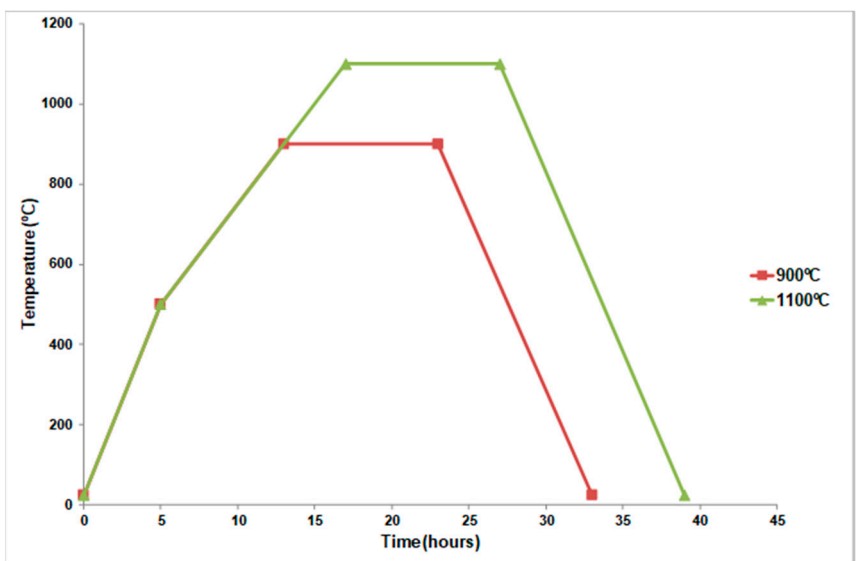

**Figure 1.** Brick heating program.

### 2.5. Testing Methods

#### 2.5.1. Materials Characterization

A D8 Advance A25 equipment from BRUKER (Bruker AXS GmbH, Karlsruhe, Germany) was utilized for the X-ray diffraction of RM (before and after treatment), and DIFFRAC-EVA software (Bruker AXS GmbH, Karlsruhe, Germany) was used for phase identification. Phase identification and accurate quantitative phase analysis (amorphous and crystalline contents) are based on the RIR (reference intensity ratio) method. The software works with multiple reference databases (ICDD PDF2/PDF4+/PDF4 Minerals/PDF4 Organics databases).

Changes in mass versus temperature were carried out with duplicate thermogravimetric analysis. A TA Instrument analyzer (Q600, TA Instruments, Barcelona, Spain) using a heating rate of 20 °C/min from 20 °C to 1100 °C in an air atmosphere [21] was used.

#### 2.5.2. Compressive Strength and Physical Tests

According to EN 772-13 [22], the weight and volume of four samples were used to calculate the material's bulk densities. According to EN 772-21 [23], the water absorption capacity was assessed in triplicate. A Tinius–Olsen machine (model TO 317, Tinius

Olsen Ltd., Surrey, England) was used to evaluate three samples of each composition's compressive strength according to EN 772-1 [24].

### 2.5.3. Environmental Assessment: Leaching Experiments

An environmental evaluation must be carried out when a secondary raw material containing heavy metals is fired to manufacture bricks. The mass loss that occurs during the heating process causes an increment of the proportion of heavy metals (which are not volatilized) in the finished brick. However, the brick's matrix is altered during the sintering process, and some heavy metals may end up stabilizing in this matrix. Since red mud retains some metals, two types of leaching tests (batch and monolithic) were tested on raw materials and final bricks, respectively, to achieve an extensive environmental characterization.

EN 12457-4 [25] is a batch static extraction test for granular materials that agitates the sample for one day at a liquid/solid ratio of 10 L/kg. This test was conducted on RM and CL. The bricks underwent a monolithic test, as specified in NEN 7375 [26], which was the other leaching test employed. A water leaching solution with a pH of 7 must be changed eight times in accordance with NEN 7375. This test simulates the effect of rain, the main leaching agent outdoors. An ICP spectrometer (Agilent Technologies, Madrid, Spain) from the Research, Technology, and Innovation Center at the University of Seville (CITIUS) was used. Two samples of each composition were used for each of the two leaching tests.

### 2.5.4. Radionuclide Activity

Particle sizes of bricks were reduced using a mill, then dust was introduced into an $80 \ cm^3$ volume polystyrene Petri dish. To prevent the escape of 222-Rn, the dish was later vacuum-sealed in a plastic bag. The gamma emissions of 214-Pb can be used to estimate the activity of 226-Ra. The gamma emissions of 40 K at 1460 keV were used to directly assess its activity, and the activity concentration of 232-Th activity was obtained from the emissions of 228-Ac.

A Canberra low-background high-purity germanium GR-6022 reverse electrode coaxial detector (Mirion Technologies (Canberra) GmbH, Hamburg, Germany) was employed as the main gamma-ray detector. It displays a 60% relative efficiency while being shielded by a 10 cm layer of high-purity lead. There were two composition measurements made. The minimum detectable activity and the decision level for 40-K, 214-Pb, and 228-Ac were calculated according to the ISO 11929-4 standard [27] at the Research, Technology, and Innovation Center at the University of Seville (CITIUS).

### 3. Results

#### 3.1. Leaching Extraction

Table 4 shows the composition of the leachates obtained when the leaching agents were used in the screening stage. The pH values of the leachate varied between 0.3 and 0.8 in the case of sulfuric acid and between 13.2 and 13.3 in the case of NaOH. When only water was used, the pH was slightly acidic (5.4).

The leaching yields when water was used as leaching agent were very low for almost all elements. Only 12% of Se was leached with water. When NaOH was used, a certain amount of V (5–7%) and Ga (3–4%) could be recovered. Se (31–72%) was also leached.

In the $H_2SO_4$ leaching tests, higher extraction yields were obtained. Ce and La were recovered from RM, achieving a yield greater than 60%. When the rest of the elements were classified according to their recovery percentage, four groups could be defined:

- Less than 10%: As, Ba, Ga, Mo, Mn, Pb, Se;
- 10–20%: Cr, Cu, Ni, Ti, and V;
- 20–50%: Sb and Zn;
- More than 50%: Ce, La, Sr, and Th.

In pH-dependent leaching behavior, elements have been classified in cationic, anionic, or amphoteric patterns [18]. The leaching behavior of the studied elements suggested an anionic pattern for Se and a cationic pattern for Ce, La, Sr, Th, Sb, and Zn. The mechanism

of leaching for most elements from RM was dissolution [18]. In the case of the element with low recovery percentage, leaching was probably controlled by the dissolution of oxides, since in the case of elements showing high recovery yields, leaching probably occurred due to the dissolution of hydroxides.

**Table 4.** Leachate composition in screening experiments.

| | Water | Sulfuric Acid | | | Sodium Hydroxide | | |
|---|---|---|---|---|---|---|---|
| | | 0.5 M | 1 M | 2 M | 0.5 M | 1 M | 2 M |
| pH | 5.4 | 0.8 | 0.4 | 0.3 | 13.1 | 13.2 | 13.3 |
| Component | | | | Metal content (ppb) | | | |
| As | <20 | 230 | 290 | 550 | 200 | 340 | 310 |
| Ba | 9 | 67 | 80 | 175 | <5 | <5 | 12 |
| Cr | 2575 | 5400 | 13,890 | 19,000 | 2324 | 2200 | 1980 |
| Cu | 14 | 850 | 960 | 1030 | <10 | <10 | <10 |
| Mn | <2 | 750 | 1347 | 1580 | <5 | <5 | <5 |
| Mo | 27 | <30 | 30 | 76 | <100 | <100 | <100 |
| Ni | <5 | 66 | 110 | 187 | <5 | <5 | <5 |
| Pb | <5 | 98 | 800 | 560 | <25 | <25 | <25 |
| Se | 36 | 38 | <30 | <10 | 90 | 210 | 120 |
| Sn | <25 | <50 | <50 | <50 | <50 | <50 | <50 |
| Sr | 7 | 5945 | 5679 | 4710 | <5 | 15 | <5 |
| Ti | <5 | 63,000 | 452,000 | 612,000 | <25 | <25 | <25 |
| V | 133 | 8260 | 16,800 | 17,900 | 7800 | 8340 | 5470 |
| Zn | 111 | 560 | 580 | 1500 | 170 | 140 | 490 |
| Sb | 31.5 | <25 | 114 | 139 | <25 | <25 | <25 |
| Ga | 18.3 | 360 | 522 | 700 | 288 | 290 | 225 |
| Ce | <25 | 12,000 | 16,700 | 17,100 | <50 | <50 | <50 |
| La | <5 | 5690 | 7220 | 7437 | <5 | <5 | 237 |
| Th | <10 | 1860 | 5170 | 6270 | <25 | <25 | <25 |

From the point of view of the valorization of RM and the recovery of valuable metals, $H_2SO_4$ seems to be the best leaching option, which is consistent with other studies [28,29]. The most valuable metals potentially recovered by $H_2SO_4$ leaching were Ni (€ 43/kg), Sn (€ 38/kg), Ga (€ 338/kg), V (€ 21/kg of $V_2O_5$), La (€ 4/kg), and Th (€ 160/kg), according to the London Metal Exchange [30] and ISE [31].

According to Borra et al. [32], the main disadvantages of RM acid leaching are the acid consumption, the low-pH effluents handling, and the difficulty in using the residual RM after leaching. However, regarding the problem of acid effluents, it should be noted that the acid leachate would be sent to a metal recovery process, which in most cases will include one or more stages of precipitation at an elevated pH, thus implying the neutralization of the effluent [33].

To reduce the negative environmental impacts, residue reutilization is analyzed in another section. In relation to the volume of effluents generated during the valorization process, in the second phase of the leaching experiments, L/S was varied from 10 to 2 to reduce effluents. Furthermore, the contact time between the RM particles and $H_2SO_4$ in the leaching stage was also studied. The results obtained using 2 M $H_2SO_4$ as the leaching agent are shown in Table 4.

### 3.2. Mathematical Leaching Models Description and Statistical Evaluation

From the data in Table 5, the extraction yields were calculated, and the results were analyzed using Design Expert. A statistical analysis was then performed to check the variation in performance of each element extraction yield with operating variables. As a result, a mathematical model was obtained that relates the leaching performance with the contact time (t) and the L/S ratio (L/S) for each element. All the models obtained were represented and can be seen in the Supplementary Materials.

**Table 5.** Leachates composition in optimization experiments.

| Contact Time | 1 h | | | 2 h | | | 6 h | | | 24 h | | |
|---|---|---|---|---|---|---|---|---|---|---|---|---|
| L/S ratio | 2 | 5 | 10 | 2 | 5 | 10 | 2 | 5 | 10 | 2 | 5 | 10 |
| As (ppb) | 700 | 680 | 500 | 756 | 570 | 500 | 785 | 507 | 500 | 1460 | 500 | 550 |
| B (ppb) | 100 | 100 | 100 | 100 | 100 | 100 | 100 | 100 | 100 | 200 | 100 | 175 |
| Cr (ppb) | 12,400 | 9920 | 5490 | 14,050 | 10,950 | 6650 | 29,128 | 12,400 | 7930 | 36,560 | 20,200 | 19,000 |
| Cu (ppb) | 1910 | 1330 | 670 | 2100 | 1290 | 690 | 2940 | 1320 | 740 | 4300 | 1540 | 1030 |
| Mn (ppb) | 1386 | 1020 | 530 | 1660 | 1130 | 650 | 4070 | 1390 | 860 | 4010 | 2220 | 1580 |
| Mo (ppb) | 100 | 100 | 100 | 100 | 100 | 100 | 100 | 100 | 100 | 100 | 100 | 76 |
| Pb (ppb) | 290 | 860 | 530 | 260 | 970 | 620 | 550 | 770 | 700 | 1700 | 1010 | 560 |
| Sr (ppb) | 9090 | 8410 | 5200 | 9790 | 7740 | 5340 | 12,109 | 7740 | 5350 | 18,530 | 8260 | 4710 |
| Ti (ppb) | 183 | 337 | 204 | 232 | 355 | 253 | 500 | 413 | 315 | 612 | 632 | 612 |
| V (ppb) | 32,160 | 27,570 | 14,860 | 32,710 | 24,920 | 15,330 | 39,100 | 25,010 | 16,160 | 60,800 | 27,110 | 17,900 |
| Zn (ppb) | 2840 | 810 | 1370 | 1780 | 710 | 410 | 2710 | 1470 | 985 | 3580 | 1750 | 1500 |
| Ce (ppb) | 19,150 | 17,470 | 10,330 | 20,520 | 16,890 | 11,100 | 30,800 | 17,290 | 11,570 | 46,090 | 20,420 | 17,100 |
| La (ppb) | 11,420 | 9340 | 5400 | 11,980 | 9090 | 5690 | 16,970 | 9180 | 5880 | 32,030 | 10,720 | 7437 |
| Th (ppb) | 6210 | 5580 | 3060 | 6620 | 5630 | 3430 | 11,290 | 5960 | 3760 | 15,370 | 7360 | 6270 |

Except in the case of the model of Ce, the mathematical models obtained were representative, with the $p$-value < 0.0001 in most cases, and < 0.001 in the models of Mn, Mo, Ti, and Th. Therefore, these models allowed for navigation in the study space. In the Supplementary Materials, the terms that should not be included in the models due to their low representativeness ($p$-value > 0.05) are marked in gray. The $R^2$ of the mathematical models that modeled the leaching of Ce, La, Pb, Th, and V were >0.75, and the rest were greater than 0.9. In the models obtained for all elements except Mn, the L/S ratio showed less influence on the extraction performance than the contact time. The maximum leaching yields of each element were reached for different combinations of operational factors (L/S ratio and leaching time) (see Supplementary Materials). The software Design Expert allows for the determination of the optimal combination that simultaneously satisfies the criteria placed on a group of responses (leaching yields).

In this case, five objectives were imposed: maximize the recovery of elements such as La (1) or Ce (2), and minimize the presence of potentially hazardous elements such as As (3), Cr (4), and Th (5). The software determined that the optimal operating conditions to satisfy the objectives must be L/S = 10 and time = 24 h (Table 6). However, although the L/S ratio increased the overall extraction yield when a larger amount of leaching agent was used, some disadvantages arose because the metal concentration in the leachates was lower. This will probably penalize the subsequent recovery stage because the consumption of reagents in the next effluent management steps increases, and this entails an economic and environmental penalty. However, although the L/S = 2 ratio achieved a leachate with a very high concentration of metals, the recovery yields were lower than those achieved when higher L/S ratios were used. Considering the reasons mentioned above and the manageability of the mixture, the L/S = 5 ratio was finally chosen. Therefore, 2 M $H_2SO_4$ was selected as the leaching solution, with a contact time of 24 h and an L/S ratio = 5 as the leaching conditions. Table 5 displays the recovery yields. As can be observed, the yields obtained experimentally and those predicted by the mathematical model were very similar. Although in this work the conditions described above were selected, using this mathematical model it is possible to estimate the leaching yields reached in any operating conditions within the study range. Under the selected conditions (L/S = 5 and 24 h), acid leaching with 2 M $H_2SO_4$ was carried out at 5 L/1 kg RM, so that there was enough solid waste to perform all the tests described below.

**Table 6.** Experimental and predicted leaching yields (%).

|  | As | Ba | Cr | Cu | Mn | Mo | Pb | Ti | V | Zn | La | Ce | Th |
|---|---|---|---|---|---|---|---|---|---|---|---|---|---|
| L/S = 10 and 24 h of leaching time | | | | | | | | | | | | | |
| Experimental | 2.5 | 0.5 | 32.0 | 7.9 | 9.7 | 1.6 | 1.8 | 2.6 | 15.3 | 32.6 | 66.0 | 56.0 | 51.5 |
| Predicted | 2.5 | - | 30.4 | 9.3 | 9.5 | 1.8 | 2.5 | 2.2 | 14.6 | 27.8 | 61.2 | 57.6 | 51.0 |
| L/S = 5 and 2 h of leaching time | | | | | | | | | | | | | |
| Experimental | 3.9 | 0.3 | 55.6 | 10.2 | 14.1 | 4.1 | 8.1 | 8.3 | 11.5 | 19.0 | 47.6 | 39.4 | 30.2 |
| Predicted | 5.5 | - | 58.6 | 11.2 | 15.2 | 3.5 | 7.2 | 10.2 | 10.6 | 20.4 | 47.7 | 41.7 | 31.3 |

### 3.3. Evaluation of the Solid Waste Remaining after Leaching

The solid waste remaining after the acid leaching attack of RM (RM-T) was dried at 105 °C for 24 h, and then its use for brick fabrication was evaluated. Figure 2 shows the mineralogical composition of the RM before and after leaching extraction and the clay. XRD data of the as-received RM were previously described by Arroyo et al. [14]. A large peak in practically all of the two ranges was visible in the XRD pattern, which is indicative of an amorphous material (82.6%). Gibbsite, hematite, and titanium–nickel oxide were the main mineralogical phases observed in RM. Compounds of iron and molybdenum were also observed. The solid obtained after acid attack (RM-T) showed a similar pattern, with practically the same mineralogical phases (iron and molybdenum compounds practically disappeared). The amorphous content of RM-T was 73.4%, resulting in a reduction of 11.1%. This was the main effect of the attack on RM according to this technique. On the contrary, clay is a less amorphous material (its amorphous content measured by XRD reached 23.8%), with αquartz, muscovite, and nontronite as the main crystalline phases [14].

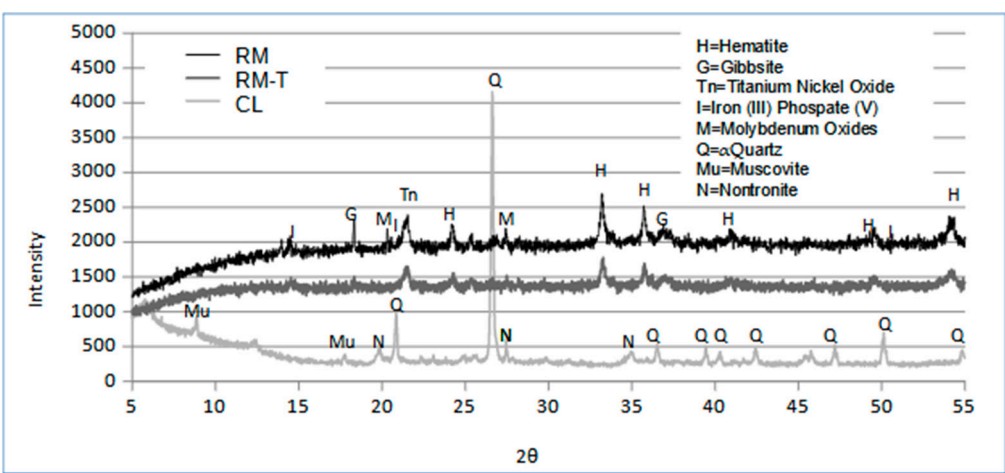

**Figure 2.** Mineralogical composition of red mud before and after the leaching process.

The particle size distribution of RM before and after the leaching attack is depicted in Figure 3. As can be seen, after leaching, particle size increased due to the agglomeration process, possibly produced during the leaching process. Clay (CL) was previously sieved prior to use, and its particle size was less than 150 μm.

Thermogravimetric analysis of RM, RM-T, and CL was carried out, and the results are shown in Figure 4. Mass loss due to moisture was observed in the range of 20–200 °C, which was higher in RM-T than in RM due to the leaching process. It is possible to see many endothermic peaks between 200 and 400 °C because of the dehydration of gibbsite to produce boehmite and alumina [2]. Between 600 and 800 °C, the decomposition of calcium carbonate is shown.

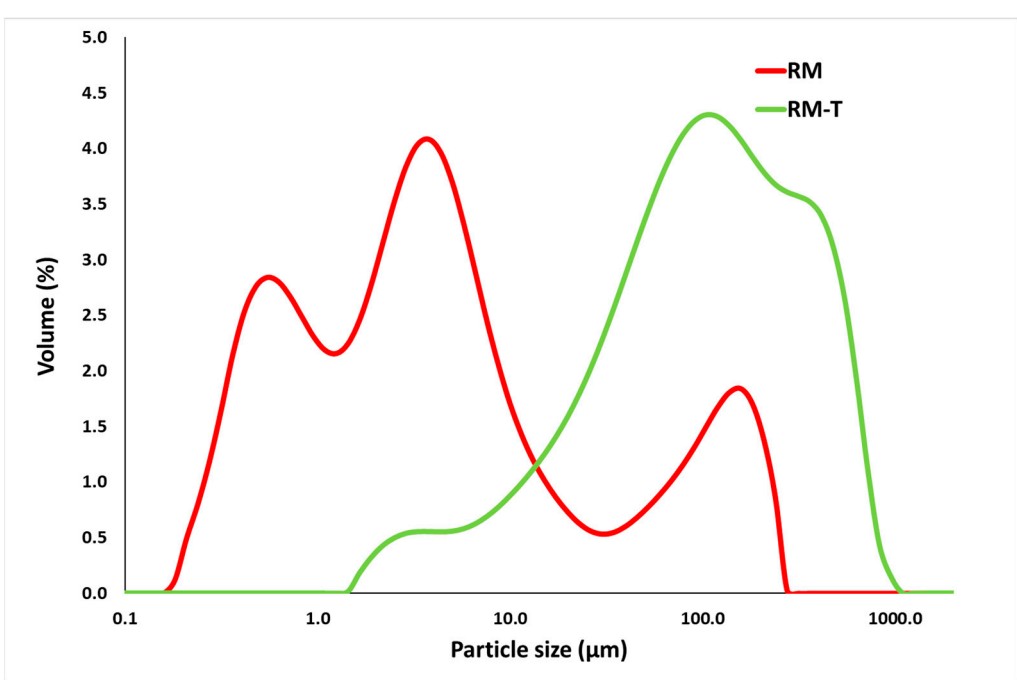

**Figure 3.** Particle size distribution of red mud before and after leaching extraction.

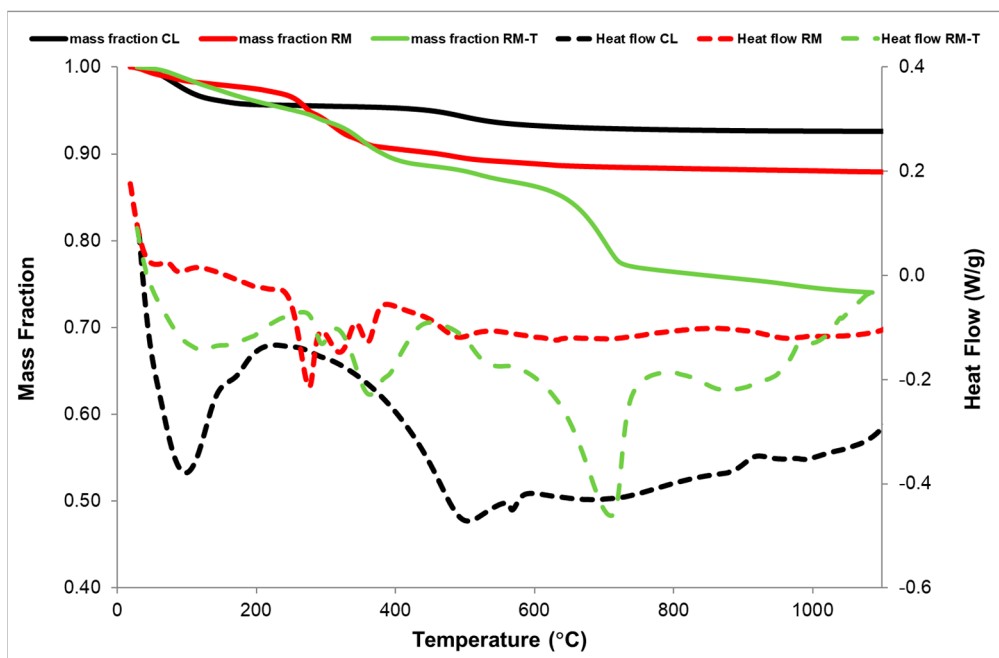

**Figure 4.** Thermogravimetric analysis of RM, RM-T, and CL.

In the case of CL, from 20 to 200 °C, a mass loss of 4% was produced, which could be attributed to the loss of moisture and physically adsorbed water. Between 60 and 160 °C, clay dehydration may be considered [34]. In the range of 200–400 °C, the weight remained almost constant. Between 400 and 800 °C, the combustion of some organic compounds [16] and the dehydroxylation of muscovite [35] could be responsible for the mass loss (3%) observed. Above 900 °C, no substantial weight changes were observed.

### 3.4. Physical Properties and Compressive Strength of Bricks

Figure 5 shows that a higher firing temperature produced a higher brick density with the same RM-T content. The flow of a viscous amorphous phase into the internal pores

was caused by the greater sintering process of RM-T and CL at higher temperatures [16]. Additionally, the disintegration of alkaline chemicals found in CL and RM-T resulted in a more compact and denser matrix at 1100 °C [36,37]. However, when RM-T was used, the density of the brick decreased compared to that measured when CL and the original RM were used to build bricks at the same temperature [14]. This effect was probably due to (a) the larger particle size of RM-T (Figure 3) and (b) the greater loss of mass produced during the heating process. Typical brick bulk densities are between 1200 and 1600 kg·m$^{-3}$ [38]. As can be seen, the RM-T compositions tested fell within that range.

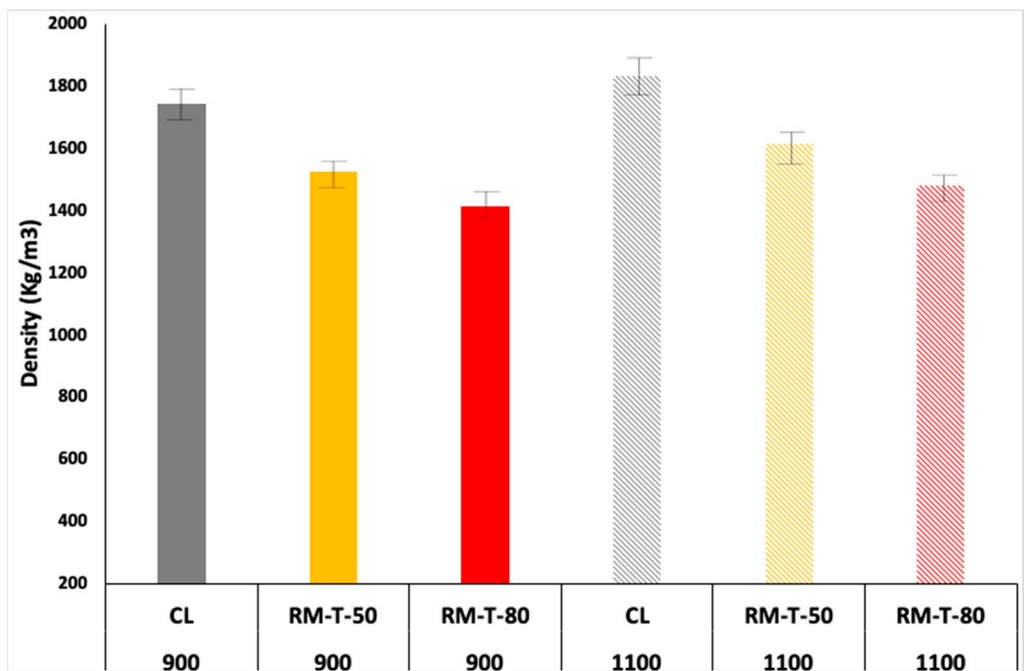

**Figure 5.** Density of samples after the sintering process.

Brick water absorption presented the opposite behavior to brick density because a lower density was produced by a higher porosity, and therefore a higher water absorption capacity, as Figure 6 shows. Water absorption should be less than 20%, per Chinese requirements [39]. As can be seen, only bricks fired at 1100 °C complied with this Chinese requirement.

The sintering process enhanced the connection between the particles, which resulted in a brick with greater mechanical strength and density, and thus raised the compressive strength in all cases when the temperature was increased (Figure 7). For the same reasons previously cited for variations in brick density, the compressive strength dropped at both temperatures when the RM-T concentration was raised.

Comparing the results of these bricks with the same composition using RM without the previous extraction process and the same sintering temperature [14], the compressive strength at 900 °C was very similar, but at 1100 °C, the compressive strength was lower when RM-T was used, probably due to the limiting effect of the previous leaching in the sintering process [40], as can be seen in the thermogravimetric analysis of Figure 4.

Compressive strength requirements for normal and moderate weathering bricks are higher than 10.3 MPa and higher than 17.2 MPa, respectively, according to ASTM C62-13 [41]. For typical bricks, the European Standard EN 771-1 for Masonry Units establishes a compressive strength higher than 10 MPa [42]. Colombian technical code NTC 4205 [43] requires a compressive strength higher than 14 MPa for non-structural bricks and 20 MPa for structural bricks. As can be seen, none of the bricks for different compositions and sintering temperature satisfied these limits, but this was due to the lower molding pressing during the fabrication process (5 MPa), when the typical ranges were between 5 and 50 MPa [14].

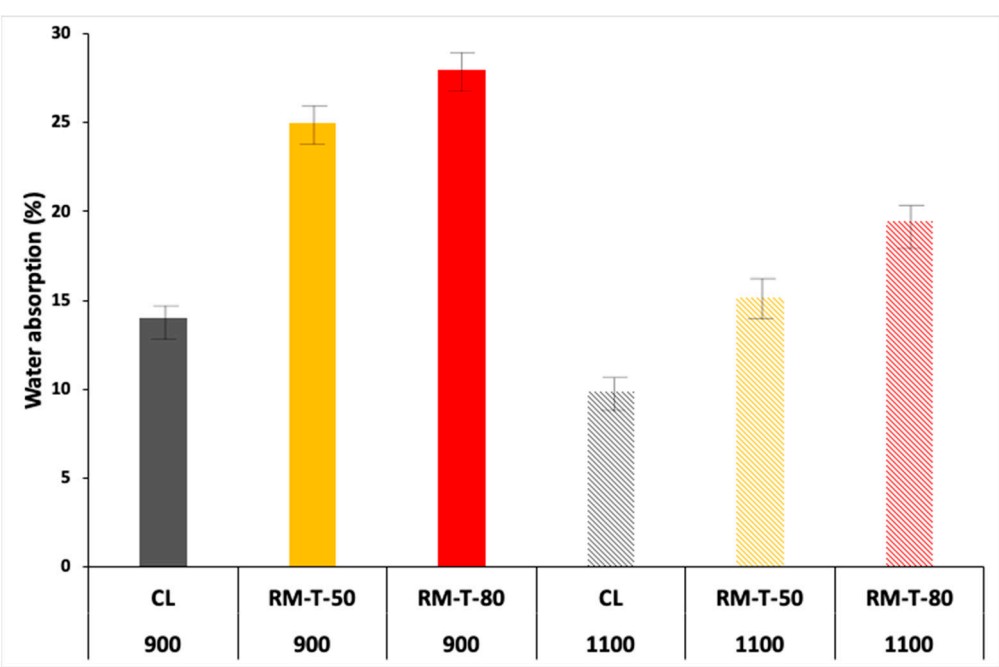

**Figure 6.** Water absorption of samples after sintering.

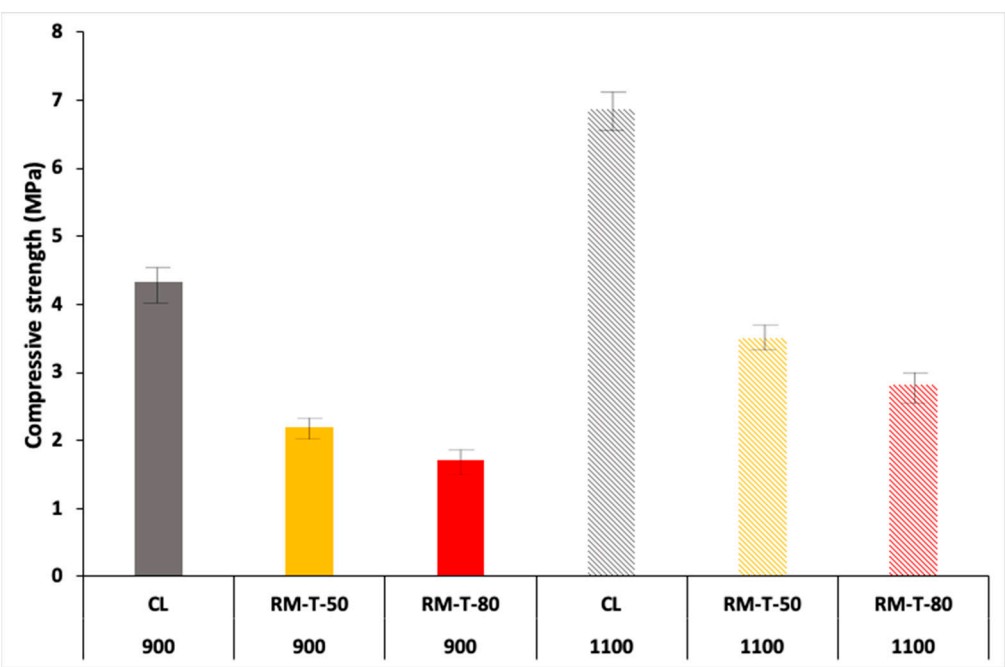

**Figure 7.** Compressive strength of the samples after sintering.

### 3.5. Environmental Properties

Table 7 shows the cumulative concentrations of heavy metals after the NEN 7375 test. As can be seen, all brick compositions met the heavy metals limits required for building materials containing wastes, according to the Dutch Soil Quality Decree [44].

The use of RM-T in bricks reduced the leaching of heavy metals when compared with the bricks prepared with the RM without any previous extraction [14], since a large part of metals was removed during the extraction. Leaching values decreased, except for As. This reduction in leaching could be due to the decrease in porosity caused by the sintering effect, which fixes heavy metals to the structure. On the contrary, the comparatively high

As leaching at 1100 °C can be explained by the high-temperature conversion of arsenide compounds into arsenates [45].

**Table 7.** Leaching of bricks according to NEN-7375 (mg/m$^2$).

|  | CL-900 | CL-1100 | RM-T-50-900 | RM-T-50-1100 | RM-T-80-900 | RM-T-80-1100 | Limits |
|---|---|---|---|---|---|---|---|
| As | 0.73 | 2.83 | 1.56 | 2.41 | 1.09 | 1.43 | 260 |
| Ba | 0.49 | 0.43 | ≤0.4 | ≤0.4 | ≤0.4 | ≤0.4 | 1500 |
| Cd | ≤0.4 | ≤0.4 | ≤0.4 | ≤0.4 | ≤0.4 | ≤0.4 | 3.8 |
| Co | ≤0.4 | ≤0.4 | ≤0.4 | ≤0.4 | ≤0.4 | ≤0.4 | 60 |
| Cr | ≤0.4 | ≤0.4 | ≤0.4 | ≤0.4 | 0.42 | ≤0.4 | 120 |
| Cu | 0.55 | ≤0.4 | ≤0.4 | ≤0.4 | ≤0.4 | ≤0.4 | 98 |
| Hg | ≤1.26 | ≤1.26 | ≤1.26 | ≤1.26 | ≤1.26 | ≤1.26 | 1.4 |
| Mo | ≤0.4 | ≤0.4 | 1.01 | ≤0.4 | 2.42 | 0.60 | 144 |
| Ni | ≤0.4 | ≤0.4 | ≤0.4 | ≤0.4 | ≤0.4 | ≤0.4 | 81 |
| Pb | ≤0.4 | ≤0.4 | ≤0.4 | ≤0.4 | ≤0.4 | ≤0.4 | 400 |
| Se | ≤0.4 | ≤0.4 | ≤0.4 | ≤0.4 | ≤0.4 | ≤0.4 | 4.8 |
| Sn | ≤0.4 | ≤0.4 | ≤0.4 | ≤0.4 | ≤0.4 | ≤0.4 | 50 |
| V | 5.29 | 0.76 | 119.46 | 6.71 | 235.48 | 36.59 | 320 |
| Zn | 1.68 | 0.48 | ≤0.4 | ≤0.4 | ≤0.4 | ≤0.4 | 800 |
| Sb | ≤0.4 | ≤0.4 | ≤0.4 | ≤0.4 | ≤0.4 | ≤0.4 | 8.7 |
| Th | ≤0.4 | ≤0.4 | ≤0.4 | ≤0.4 | ≤0.4 | ≤0.4 | - |

Radionuclides are present in natural building materials and in some by-products. The 2013/59/EURATOM Directive [46] establishes the standards for the radiological impact-aware recycling of wastes and byproducts into building materials. The Directive establishes a maximum gamma radiation dose of 1.0 mSv/y for prolonged environmental exposure to natural radiation to achieve this goal.

For this assessment, the activity concentration index (ACI) was used. Equation (1) states that this value relies on the activity concentrations of the principal natural radionuclides, K-40, Th-232, and Ra-226.

$$ACI = (CTh\text{-}232/200) + (CRa\text{-}226/300) + (CK\text{-}40/3000) \tag{1}$$

where $C_{Th\text{-}232}$, $C_{Ra\text{-}226}$, and $C_{K\text{-}40}$ (in Bq/kg) are the activity concentrations of Th-232, Ra-226, and K-40, respectively. A building material must have an ACI lower than 1.0 to comply with the yearly dose threshold of 1.0 mSv/y (Table 8).

**Table 8.** Radionuclide activity concentrations (Bq/kg) and activity concentration indexes (ACI).

| | Materials | | | |
|---|---|---|---|---|
| Radionuclides | RM | CL | CL-1100 | RM-T-80-1100 |
| K-40 | 98 | 650 | 618 | 203 |
| Ra-226 | 235 | 35 | 11 | 4 |
| Th-232 | 249 | 37 | 33 | 104 |
| ACI | 2.06 | 0.52 | 0.41 | 0.6 |

As can be seen, the main concentration of natural radionuclide activity in the red mud studied was due to Th (higher concentration and higher coefficient in Equation (1)). The amount of Th lost in the previous hydrometallurgical process was estimated to be 51.48%, so the emission of Th in the brick was much lower when RM-T was used, meeting the ACI < 1 limit. On the contrary, when RM was used as raw material to manufacture bricks, employing the same proportions and the same firing temperatures, the $C_{Th\text{-}234}$ measured was 186 Bq/kg [14] (that can be compared with the value observed in the case of the RM-T bricks: 104 Bq/kg), exceeding the ACI limit.

## 4. Conclusions

Red mud is a complex by-product of industry. It contains not only several valuable elements but also hazardous elements, including radionuclides. For this reason, RM valorization is a challenge that requires a specific experimental strategy. This work is framed within the circular economy, firstly recovering the valuable metals of the red mud and secondly recycling the treated red mud as a component of fired bricks, analyzing the changes of the properties of the waste produced by the recovery process.

In this paper, RM valorization was studied in a holistic way. First, hydrometallurgical methods were used for metal extraction; second, the remaining solids were used as raw materials for brick manufacturing. From an environmental point of view, the effluents generated by the leaching of the water were slightly acidic but did not allow for the recovery of most valuable elements. On the other hand, leaching using $H_2SO_4$ or NaOH was more effective, but effluents needed a pH adjustment of effluents before discharge.

Regarding $H_2SO_4$ leachates, the pH adjustment should be performed after metal recovery, which is usually achieved in an alkaline medium. Considering technical, economic, and environmental points of view, the optimal conditions to recover valuable elements from RM imply the use of 2 M $H_2SO_4$ in contact with RM for 24 h at an L/S ratio of 5. In this way, high leaching yields of valuable elements, some of which are considered critical raw materials by the EU, such as La (47.6%) or V (11%), can be achieved.

When residual red mud after extraction is used as the brick component, the mechanical properties of the bricks decrease as their RM percentage is increased, and the residual red mud presents slightly worst characteristics than untreated red mud as a component of fired bricks.

Regarding the environmental characteristics studied, heavy metals leaching and the Activity Concentration Index decrease in the bricks containing treated red mud compared with bricks prepared with untreated RM and are similar to those prepared with natural clay.

**Supplementary Materials:** The following supporting information can be downloaded at: https://www.mdpi.com/article/10.3390/pr10112367/s1, Figure S1. Metal leaching yields: mathematical model and response surfaces. Table S1. Statistical analyses of metal leaching yields: mathematical models.

**Author Contributions:** Conceptualization, C.L. and F.A.-T.; methodology, Y.L-G.; software, F.A.-T.; validation, R.V., L.F.V., and C.F.P.; formal analysis, C.F.P.; investigation, C.L., F.A.-T., and Y.L.-G.; resources, C.F.P.; data curation, R.V.; writing—original draft preparation, L.F.V.; writing—review and editing, C.L.; visualization, F.A.-T.; supervision, C.L.; project administration, C.F.P.; funding acquisition, L.F.V. All authors have read and agreed to the published version of the manuscript.

**Funding:** This research was funded by financial support for this study provided by the Junta de Andalucía through the project "Integración de Tecnologías Emergentes de Membranas para la Valorización de Efluentes de la Industria Minero-Metalúrgica" (PAIDI 2020).

**Data Availability Statement:** Not applicable.

**Acknowledgments:** The authors would like to thank Manuel Valenzuela Mateo, Mª Luisa Martínez Domínguez, and Miguel Rodríguez Carrillo their technical support.

**Conflicts of Interest:** The authors declare no conflict of interest.

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
