# Peer review of "Valorization of Bayer Red Mud in a Circular Economy Process: Valuable Metals Recovery and Further Brick Manufacture"

_processes, doi:10.3390/pr10112367_

Round 1

Reviewer 1 Report

the topic is original using waste materials for fired clay bricks. however, some points should be improved before acceptance.

if possible the authors should carry out the sem analysis and XRD of hardened samples for more explanation. as the description is not enough for publication. 

for Figures 3 and 4 please change the comma to point on the y-axis 

the authors must compare their work with standard fired clay bricks with or without chemical reagents as reported here

https://doi.org/10.1016/j.jeurceramsoc.2020.01.035

https://doi.org/10.3390/ma14112903

please rewrite the conclusion please shortens the sentences

Author Response

Thank you for your comments.

The paper has been improved. We have considered reviewers questions and English grammar has been revised.

Reviewer 2 Report

This research explored a process of treating red mud by, sulfuric acid and sodium hydroxide. It is not creative because these routes are very common. For the leachate, the authors did not propose a feasible method to separate the metals in the leaching solution. As we know, the separation of too much metals  in the solution is very difficult. And the the solid remaining was used for brick production, as is also very common. So it is suggested to reject this manuscript.   

Author Response

(The authors gave the same response as above.)

Reviewer 3 Report

Dear authors,

Please see my comments and suggestions below:

1. Abstract
Indicate the actual doses of red mud and sintering temperature, proportion of of red mud. Are you referring to compressive strength as the mechanical properties in line 18?

2. Introduction
Adequately present how your study is align to circular economy.

3. Materials and Methods
Indicate the technique/s used for complete chemical characterization of red mud.
Explain why 0.5M, 1M and 2M of sulfuric acid and sodium hydroxide were used.
Indicate the actual experimental design and statistical analyses employed
What's your basis for choosing the employed heating procedure?
Indicate the technique used to determine the % amorphous in line 229
Was compressive strength the mechanical properties measured?
Indicate your reference for the Radionuclide activity (standard?) test 

4. Results
Use relevant mechanism/s of leaching in acidic and basic medium to further explain your results
Indicate important results such as test statistic and p-values in presenting and discussing your results.
Identify other XRD peaks 
Are the measured properties acceptable for brick application?

5. Conclusion
Elaborate the basis of lines 348-353
Revise the conclusion to better present the circular economy perspective of the current study 

Author Response

(The authors gave the same response as above.)

Round 2

Reviewer 1 Report

all the recommendations are well addressed by the authors accordingly

I recommend this paper for publication

Author Response

Thank you for your comments

Reviewer 3 Report

Thank you for sending your revisions.
Your manuscript has dramatically improved, however, there are some comments that are not addressed satisfactorily as follows:

1. Abstract

Indicate the actual doses of red mud (or proportion of red mud) and sintering temperature.
Red mud doses or proportion of red mud, clay and water is not stated in section 2.3 Brick manufacture. section 3.2.1 should be in the methodology

2. make a separate section for the design of experiment and statistical analysis employed

3. Add in the methodology the employed technique (relevant concepts or standard procedure or formula, not just the software) to compute the % amorphous phase of your samples

4. Discuss your procedure to determine the optimal conditions to recover valuable elements from RM

5. Use a period instead of a comma for decimal values (both in the manuscript and supplementary file)

6. Properly label, format and organize your figures and tables (both in the manuscript and supplementary file)

7. Please correct the decimal point of the P-value of leaching on page 5 of the supplementary file

8. Review your manuscript for grammatical and formatting errors (such as chemical formula)

9. Revise the title of section 3.3

Author Response

Authors answer: Thank you for your kind comments. 

1. Abstract 

Indicate the actual doses of red mud (or proportion of red mud) and sintering temperature. 

Authors answer:  

The abstract has been modified 

“Two doses of red mud (50 and 80%p) and two different sintering temperatures (900 and 1100ºC) were tested.” Red mud doses or proportion of red mud, clay and water is not stated in section 2.3 Brick manufacture. section 3.2.1 should be in the methodology 

This content has been moved to section 2.3 Brick Manufacture: 

Table 3 shows the proportions of RM-T and CL, and the water / solid ratio. Two firing temperatures (900 and 1100oC) were tested.  

Table 3. Brick compositions (wt %)  

CL  

RM-T  

Water/solid ratio  

CL-900  

100  

0  

0.12  

RM-T-50-900  

50  

50  

0.30  

RM-T-80-900  

20  

80  

0.35  

CL-1100  

100  

0  

0.12  

RM-T-50-1100  

50  

50  

0.30  

RM-T-80-1100  

20  

80  

0.35  

As can be seen in Table 3, the water/solid ratio was higher when the RM-T content increased to obtain a fluid mixture. 

  1. make a separate section for the design of experiment and statistical analysis employed

Authors answer: A new section has been included (see new section 3.2) 

  1. Add in the methodology the employed technique (relevant concepts or standard procedure or formula, not just the software) to compute the % amorphous phase of your samples

Authors answer:  Phase identification and accurate quantitative phase analysis (amorphous and crystalline contents) are based on RIR (reference intensity ratio) method. Software works with multiple reference databases (ICDD PDF2/PDF4+/PDF4 Minerals/PDF4 Organics databases).

  1. Discuss your procedure to determine the optimal conditions to recover valuable elements from RM

Authors answer: A new discussion has been included in section 3.2 

  1. Use a period instead of a comma for decimal values (both in the manuscript and supplementary file)

Authors answer: Commas have been changed by dots in decimal values 

  1. Properly label, format and organize your figures and tables (both in the manuscript and supplementary file)

Authors answer: Figure and Table organization and format have been revised 

 7. Please correct the decimal point of the P-value of leaching on page 5 of the supplementary file

Authors answer: Dot has been included in p-value 

  1. Review your manuscript for grammatical and formatting errors (such as chemical formula)

Authors answer: Manuscript has been revised 

  1. Revise the title of section 3.3

Authors answer: The old title 3.3. Physical and Compressive strength has been corrected. The new title is: 3.4. Physical Properties and Compressive strength of bricks 
